# Targeting β2-Adrenergic Receptors Shows Therapeutical Benefits in Clear Cell Renal Cell Carcinoma from Von Hippel–Lindau Disease

**DOI:** 10.3390/jcm9092740

**Published:** 2020-08-25

**Authors:** Virginia Albiñana, Eunate Gallardo-Vara, Isabel de Rojas-P, Lucia Recio-Poveda, Tania Aguado, Ana Canto-Cano, Daniel T. Aguirre, Marcelo M. Serra, Pilar González-Peramato, Luis Martínez-Piñeiro, Angel M. Cuesta, Luisa Maria Botella

**Affiliations:** 1Centro de Investigaciones Biológicas Margarita Salas, Consejo Superior de Investigaciones Científicas (CSIC), 9 Ramiro de Maeztu Street, 28050 Madrid, Spain; vir_albi_di@yahoo.es (V.A.); eunate.gallardo@yale.edu (E.G.-V.); iderojas@ucm.es (I.d.R.-P.); lrecio@cib.csic.es (L.R.-P.); tania.aguado@gmail.com (T.A.); anadelcantocano@gmail.com (A.C.-C.); 2Centro de Investigación Biomédica en Red de Enfermedades Raras (CIBERER), Group U707, 3-5 Monforte de Lemos Avenue, 28029 Madrid, Spain; 3Department of Neurosurgery, Hospital Universitario Fundación Jiménez Díaz (IIS-FJD), 2 Reyes Católicos Avenue, 28040 Madrid, Spain; dtaguirre@fjd.es; 4Department of Internal Medicine, Hospital Italiano de Buenos Aires (HIBA), 4190 Pres. Tte. Gral. Juan Domingo Perón Avenue, C1199 Ciudad Autónoma de Buenos Aires, Argentina; marcelo.serra@hospitalitaliano.org.ar; 5Department of Pathology, Instituto de Investigación IdiPaz. 6 Pedro Rico Street, 28029 Madrid, Spain; mpilar.gonzalezperamato@salud.madrid.org; 6Department of Urology, Instituto de Investigación IdiPaz, 6 Pedro Rico Street, 28029 Madrid, Spain; luis.mpineiro@salud.madrid.org

**Keywords:** β-adrenergic receptor antagonist, ICI-118,551, propranolol, HIF, ccRCC, VHL, anticarcinogenic

## Abstract

Von Hippel–Lindau (VHL), is a rare autosomal dominant inherited cancer in which the lack of VHL protein triggers the development of multisystemic tumors such us retinal hemangioblastomas (HB), CNS-HB, and clear cell renal cell carcinoma (ccRCC). ccRCC ranks third in terms of incidence and first in cause of death. Standard systemic therapies for VHL-ccRCC have shown limited response, with recurrent surgeries being the only effective treatment. Targeting of β2-adrenergic receptor (ADRB) has shown therapeutic antitumor benefits on VHL-retinal HB (clinical trial) and VHL-CNS HB (in vitro). Therefore, the in vitro and in vivo antitumor benefits of propranolol (ADRB-1,2 antagonist) and ICI-118,551 (ADRB-2 antagonist) on VHL^−/−^ ccRCC primary cultures and 786-O tumor cell lines have been addressed. Propranolol and ICI-118,551 activated apoptosis inhibited gene and protein expression of HIF-2α, CAIX, and VEGF, and impaired partially the nuclear internalization of HIF-2α and NFĸB/p65. Moreover, propranolol and ICI-118,551 reduced tumor growth on two in vivo xenografts. Finally, ccRCC patients receiving propranolol as off-label treatment have shown a positive therapeutic response for two years on average. In summary, propranolol and ICI-118,551 have shown antitumor benefits in VHL-derived ccRCC, and since ccRCCs comprise 63% of the total RCCs, targeting ADRB2 becomes a promising drug for VHL and other non-VHL tumors.

## 1. Introduction

Renal cell carcinoma (RCC) ranks as the seventh most common cancer worldwide [1]. The most common histological subtype (63%) is the clear cell renal cell carcinoma (ccRCC). Although in regular population they are sporadic, these tumors are inherited by Von Hippel–Lindau (VHL) patients and 60% of them may suffer from ccRCC in addition to other types of multisystemic tumors such as hemangioblastomas (HB).

VHL is a rare type of cancer disease with an estimated annual birth incidence of 1/36,000 and a prevalence at 1/53,000 in the general population [2,3,4]. VHL is an autosomal dominantly inherited genetic disorder and its clinical manifestations include multiple tumors that appear throughout the life span of the patient [5]. The course of VHL is associated with the development of multiple bilateral and multifocal tumors such as retinal and Central Nervous System HB (CNS-HB), ccRCC, pheochromocytomas, pancreatic islet tumors, endolymphatic sac tumors, renal and pancreatic cystadenomas, as well as broad ligament of the uterus and epididymal cystadenomas. Although VHL tumors are usually benign, RCC, pheochromocytoma, and neuroendocrine pancreatic are considered malignant tumors [6,7,8].

Kidneys of VHL patients usually feature both preneoplastic renal cysts and ccRCC [9,10]. It has been shown that the epithelial cells lining of these cysts present early loss of *VHL* and, consequently, chronic accumulation of the hypoxia inducible factor-2α (HIF-2α), the most prominent HIF in RCC. This HIF-2α accumulation drives the activation of its target genes and tumor growth [11].

In the kidney, VHL is presumed as an early tumor suppressor gene where genetic and epigenetic events (either by mutation, loss of heterozygosity or hypermethylation of the promoter) are required for tumorigenesis and it has been documented in up to 80% of the sporadic ccRCC [12]. A recent comparison of genomic profiles revealed that VHL-associated tumors showed similar copy number changes as sporadic ccRCC [13]. Since VHL lacks an effective treatment, new drugs able to prevent repeated surgeries and to delay tumor development, including ccRCC, are in demand. Most of the systemic therapies for VHL-tumors target tyrosine kinase receptors (TKI) or proangiogenic agents, but the trials for VHL have shown limited responses (Table 1) [14,15].

To isolate and culture primary HB-tumors derived from VHL patient surgeries [16,17] open a key opportunity to characterize VHL-derived HB cells and to test drugs with therapeutic aims, thus reducing the need of recurrent surgeries in VHL patients. In this way, we also have demonstrated the therapeutic properties of propranolol, an unspecific antagonist of β-1,2-adrenergic receptor (ADRB-1,2). Propranolol was able to reduce proliferation by triggering apoptosis and, in addition, to reduce the expression of HIF target genes such as *VEGF* [16,18,19].

Furthermore, a phase III clinical trial (EudraCT: 2014-003671-30) was conducted to address the safety and effectiveness of propranolol in VHL patients with retinal HB (Table 1). Briefly, the number and size of retinal HB remained stable in all patients and VEGF and miR210 plasma levels were reduced to normal levels after propranolol treatment [19,20]. Nevertheless, hypotension and bradycardia were the main side effects registered, both as consequence of propranolol’s main attribute—reducing blood pressure. Thus, it proves to be inappropriate for patients with normal or low blood pressure. These side effects of propranolol are driven exclusively by its ADRB-1 blockade, and therefore could be avoided, while keeping the previously mentioned benefits, by using a high specific ADRB-2 antagonist.

Among the β-blockers, one selective antagonist of the ADRB-2, the erythro-D,L-1(methylinden-4-xyloxy)-3-isopropylaminobutan-2-ol, known as ICI-118,551 (ICI), stands out. Cuesta et al. recently reported the similarities of ICI and propranolol effects on VHL patient-derived HB cultures, such as a specific decrease of cell viability and apoptosis triggering in CNS-HB primary tumors [18]. Based on these data, we wondered whether β-antagonists would also show its therapeutic properties on a more malignant VHL-related carcinoma, such as the ccRCC. Therefore, in line with the previous results obtained on HB, we isolated, established, and tested both β-antagonists on RCC primary tumors from VHL patients.

Since VHL-derived ccRCC grow under HIF transcriptional stimulation, as it is the case in HB, development of primary tumors from ccRCC derived from VHL-RCC surgeries was the first aim of the present work. In addition, cell viability, apoptosis, gene expression, with emphasis in *HIF-2α* and its targets, and the in vitro and in vivo carcinoma cell responses to ADRB antagonists, including propranolol and ICI, were evaluated, demonstrating their positive effects. These results were further reinforced with clinical data collected from VHL-ccRCC patients under propranolol treatment for more than 15 months, who showed a better outcome during propranolol treatment.

## 2. Results

### 2.1. Isolation and Cultivation of Primary ccRCC Tumor Cultures from Surgical Specimens

Up to 15 different VHL-ccRCC primary cultures have been collected, isolated, and cultivated, and an average of three different ccRCC samples were used for the in vitro assays (ccRCC samples were identified with an internal numerical code, indicating individual patient tissue isolates). Among them and due to cell availability, 5 different ccRCC samples were assayed for the present manuscript. Representative images from first passages of primary VHL-ccRCC tumor cultures are shown in Figure 1A. Until the first confluence, cells exhibited epithelioid morphology, growing in groups or “foci” with an elongated or rounded shape before becoming confluent cells. The already described lipid accumulation, typical for ccRCC, could be observed during the first passages of the primary culture.

The nature of the primary cultures as ccRCC is based on clinical and morphological data. First, they are derived from tumoral surgery surplus of patients previously diagnosed as suffering from VHL. Second, the preliminary clinical pathology analysis determined the kind of RCC, being classified as ccRCCs. Moreover, they have been extensively described in literature, becoming a good model to recapitulate the ccRCC [35]. In addition, the expression of the carbonic anhydrase IX (CAIX), a well-known marker for ccRCC, was tested by immunofluorescence and qPCR proving the ccRCCs identity of our primary cultures (Figure 2D and Appendix A, respectively).

VHL-derived ccRCC primary tumors are a model that may represent the ccRCC biology and which could prospectively be used for in vitro functional analyses [35]. Thus, we used these primary tumors to assess the putative therapeutic properties of β-antagonists, the non-specific propranolol and the β2-specific ICI.

### 2.2. Cell Viability of ccRCC Primary Tumors and RCC 786-O Cell Line Is Impaired after ADRB Blockers Treatment by Triggering Apoptosis

Figure 1B shows the viability in two different ccRCC primary tumors (ccRCC 4 and 7), the *VHL*^−/−^ human RCC cell line 786-O, and the *VHL^+/+^* Human Umbilical Vein Endothelial Cell (HUVECs) after treatment with different doses of propranolol or ICI for 72 h. Propranolol and ICI decrease ccRCC cell viability in a dose-dependent manner, reaching IC50 levels around 100 µM for both β-blockers. 786-O cells are proved to be more sensitive to β-antagonists, reaching less than 60% of viability at 50 µM and 40% or less at 100 µM propranolol or ICI. In addition, the ratio shown in Figure 1C indicates the number of live cells in which the apoptotic mechanisms, caspases 3/7, are activated. HUVECs, corresponding to primary endothelial VHL positive cells was used as control of healthy vasculature.

In addition, to show specificity of the ADRB blockers on tumoral cells vs. non-cancerous cells, the LD50 calculations of the ccRCC cells were done as well as previously published data from HUVECs, a non-cancerous endothelial primary culture [18], and the human renal cancer cell line 786-O. These results demonstrate that LD50 for 786-O cells ranges around 67 µM for propranolol and 55 µM for ICI, LD50 for primary ccRCC cultures range from 110 to 120 µM for both ADRB antagonists, and LD50 for HUVECs ranges around 175 µM for propranolol and 138 µM for ICI, (Appendix A). Therefore, there is a notable difference between the LD50 of ccRCC tumor cells as compared to healthy ones, in vitro, opening an important therapeutic window of opportunity.

The effect on cell death becomes more clearly visible when primary tumors are cultivated in vitro in the absence or presence of β-blockers for 72 h. Figure 1D shows a confluent cell monolayer in the absence of any treatment, and discontinuities of this layer become apparent after treatment with 100 µM propranolol or ICI. Moreover, the cellular morphology varies, becoming elongated and showing cell debris as indicative of apoptosis. Apoptosis, as origin of the decreased viability after ADRB blockers propranolol and ICI treatment on VHL-derived primary tumors, has been previously reported in literature [16,18,19].

To confirm the apoptotic cell death induction in our primary VHL-ccRCC isolates, we assessed the expression of two pro-apoptotic genes, *BAX* and *CASP9*, by quantitative PCR. Figure 1E shows that the expression of *CASP9* was significantly upregulated after 72 h of treatment with 100 µM of either propranolol or ICI (*p* = 0.00062 and *p* = 0.0020, respectively). *BAX* expression was significantly upregulated too (propranolol; *p* = 0.00013 and ICI; *p* = 0.0017).

Figure 1F reinforces the previous results shown in this figure, where the condensation and fragmentation of nuclei can be observed due to the apoptosis process in ccRCC cultures after β-blocker treatment. Taken together, propranolol and ICI seem to act in a dose-dependent manner, triggering apoptosis on both primary tumors and RCC cell lines.

Altogether, the data displayed on Figure 1C–F support the idea of apoptosis being the mechanism causing the decrease in primary ccRCC tumor cell viability after β-antagonist treatment in ccRCC.

### 2.3. Propranolol and ICI-118,551 Decrease HIF, VEGF, and CAIX Expression

Due to the lack of functional pVHL in VHL-ccRCC primary tumor cells and in 786-O cells, HIF proteins are not targeted for proteasomal degradation and can accumulate in the cell [2,3,4,5]. As shown in Figure 2A, the levels of protein HIF-2α significantly decreased after 72 h of 100 µM of either propranolol or ICI treatment. 786-O cells are more sensitive to β-blockers than ccRCC primary tumors, with propranolol being more effective in the cell line and ICI on the ccRCC primary tumors.

The amount of VEGF, a direct HIF target, secreted by 3 different primary cultures of VHL-ccRCCs and 786-O cells was significantly reduced after a 48-h treatment with 100 µM propranolol or ICI. The amount of VEGF secreted into the culture medium was measured by ELISA and compared to untreated cells. As shown in Figure 2B, propranolol (*p* = 0.0059) and ICI (*p* = 0.0005) significantly decreased the amount of VEGF secreted by 786-O cells. This was also observed in ccRCC with both propranolol (*p* = 0.014) and ICI (*p* = 0.0028), at variance with the results published by Shepard et al. [36], in which VEGF levels remained unchanged after 200 µM of propranolol treatment. Moreover, we also found a significant reduction of almost 20% in VEGF expression compared to the control, in three independent primary VHL-ccRCCs cultures treated with 100 µM of either propranolol or ICI. Figure 2C represents the mean of 3 different assays done with primary tumors obtained from 3 non-related VHL-ccRCCs. Additionally, *AQUAPORIN-1 (AQP-1)*, another HIF target highly expressed in different tumors [37,38,39], was significantly decreased after treatment with propranolol or ICI (*p* = 0.00012 and *p* = 0.00015, respectively). Interestingly and in agreement with the result observed in CNS-HBs [18], *ADRB-2* expression was also significantly downregulated by the ADRB antagonists, as shown in Figure 2C (*p* = 0.003 for propranolol and *p* = 0.021 for ICI).

Lastly, the biomarker for RCC CAIX is a HIF target. Knowing that HIF levels were reduced after ADRB antagonist treatment, we wondered about CAIX expression levels. As shown in Figure 2D, the cell line 786-O and three different VHL-ccRCCs primary cultures were treated either with PBS, propranolol, or ICI. As observed, the blockage of ADBR-2 significantly decreased CAIX expression in all the cells tested. Interestingly, HIF-2α and CAIX levels were similarly reduced in 786-O and ccRCC7 cells (Figure 2A,D). Altogether, the reduction of *HIF* expression by propranolol and ICI impairs the HIF signaling pathway in both, 786-O cell line and primary tumors, suggesting potent antitumor properties.

Figure 2 shows that the intensity of the cell response to the treatment with propranolol and ICI, is variable, and it reflects the heterogeneity of primary cultures derived from tumors of different VHL patients and harboring distinct *VHL* mutations.

### 2.4. HIF-2α Cell Distribution Is Altered by the ADRB Antagonists

Lack of VHL protein prevents HIF degradation, allowing its accumulation even under normoxic conditions (pseudo-hypoxic state), therefore triggering their downstream signaling pathways. We have observed that propranolol and ICI reduce but do not abolish HIF protein levels and impair the expression of HIF-target genes like *VEGF* and *AQP-1*. Thus, we wondered whether the ADRB antagonists propranolol and ICI could play a role in the cellular distribution of HIF. To address this question, the RCC cell line 786-O, a standard model for *VHL^−/−^*, was used to assess HIF-2α distribution in the cellular compartments (nucleus vs. cytoplasm), before and after the treatment with β-blockers, by an immunofluorescence confocal microscopy assay.

Quantitative analysis of HIF-2α distribution showed that it was more abundant in the cytoplasm (70%) than in the nuclei (30%) of untreated 786-O cells (Figure 3). When cells were treated with either propranolol or ICI, nuclear HIF-2α protein was significantly reduced to 22.7% and 23.5% (*p* = 0.0052 and *p* = 0.0195), respectively. No significant differences exist between propranolol or ICI treated cells on HIF-2α nuclear localization (*p* = 0.32970).

In summary, these data demonstrate that the β2-adrenergic antagonists propranolol and ICI, do not only decrease HIF expression but also change HIF subcellular localization. Decreased HIF localization in the nuclear compartment after β2 adrenergic antagonists treatment (Figure 3) may at least partially explain the reduced levels of *AQP-1* and *VEGF* mRNA shown in Figure 2B,C. Similar results have been obtained by our group in VHL CNS-HBs and HUVECs under hypoxic conditions [16,19] and by the group of Zhang et al. in 2016 with vorinostat, a histone deacetylase inhibitor [40].

### 2.5. NFκB-p65 Pathway Is Targeted by the ADRB Blockers

NFκB pathway is a main inflammatory pathway naturally inhibited by IκB. Upstream activating signals (e.g., binding of TNF-α, IL-1α, LPS, CD40, or other unknown ligands to its receptor) cause phosphorylation of IκB by IκB kinase (IκK). This triggers the degradation of IκB through the ubiquitin system. The free unbound NFκB protein, p65, can then translocate to the nucleus and activate transcription of target genes [41]. Malec et al. reported that 786-O cells showed a 4-fold increased NFκB activity under basal conditions, as compared to 786-O expressing *VHL* after a stable transfection, and they concluded that enhanced activity of the NFκB pathway was associated with the loss of *VHL* in these cells [42]. In line with these observations, we explored the expression and translocation of p65/RelA, a main component of the NFκB pathway, in 786-O cells in basal conditions or after treatment with either propranolol or ICI. Figure 4A show the immunofluorescence staining of p65 in control conditions and after the treatment with β-blockers. Moreover, histograms indicate the relative amount of p65 localized in the nucleus and in the cytoplasm. Under control conditions, p65 is highly expressed and almost evenly distributed between the nuclei and the cytoplasm (45% and 55%, respectively). However, β-antagonists significantly alter p65 nuclear localization, resulting in a 32% and 43% reduction in nuclear localization after propranolol (*p* = 9.13 × 10^−8^) and ICI (*p* = 6.35 × 10^−7^) treatment, respectively. Therefore, this result seems to indicate that NFkB signaling in 786-O cells is also decreased by β-blockers. Next, we wondered about the downstream effects of the β-antagonists in the NFκB signaling pathway. We tested three different ccRCCs primary tumors and analyzed the expression of three interleukins triggered by p65: IL-1β, IL-6, and TNFAIP6. As shown in Figure 4B, all these NFκB targets were significantly downregulated by propranolol and ICI.

### 2.6. ICI-118,551 and Propranolol Treatment Decreases the Growth Rate of ccRCC VHL^−/−^ 786-O Tumor Xenograft in NSG Mice

Our previous results pointed towards a potential benefit of the β-blockers for the treatment of VHL, but a confirmation with a relevant in vivo animal model was required to support this hypothesis. To achieve this, two different approaches were developed in order to mimic two scenarios: a classic solid tumor case and a VHL case.

The first in vivo approach mimics a VHL scenario, where VHL patients are periodically evaluated and their established RCCs (and/or retinal or CNS-HBs) are imaged to calculate their volume and growth speed. Typically for VHL patients in the clinic, when renal tumors increase by 3 cm in diameter, partial or total nephrectomy is required. In our in vivo study, treatment with either of the drugs (10 mg/kg/day) started on the day after the 786-O tumor cells were subcutaneously implanted (10^6^ cells/mouse) and continued for the first five consecutive days. As shown in Figure 5A, the treatment with propranolol or ICI for only five days was effective enough to significantly delay the tumor growth, especially in the case of propranolol (*p* = 0.0131 for propranolol and *p* = 0.0490 for ICI). Particularly, propranolol-treated tumors needed about 10 days more than the vehicle group to reach 100 mm^3^ (37 vs. 27, respectively). In the vehicle-treated group, tumors needed 31 days to reach about 150 mm^3^, while ICI-treated tumors needed 40 days and propranolol-treated ones 46 days. Furthermore, at the end of the experiment (day 46), average tumor size in the control group was almost double of that in the propranolol cohort (323.8 vs. 162.9 mm^3^, respectively), and approximately 35% larger in volume than ICI-treated tumors (323.8 vs. 211.9 mm^3^, respectively). Moreover, no adverse effects were observed during the whole time of the in vivo study, followed up by animal behavior observation and weight evolution.

In the second study, we tried to mimic the classical in vivo cancer therapeutic intervention. For this, the ccRCC human cell line 786-O was subcutaneously implanted into NSG mice (10^6^ 786-O cells/mouse) and propranolol or ICI treatment (10 mg/kg/day) was initiated when the tumor sizes averaged 100 mm^3^. Tumor growth kinetics are shown in Figure 5B. Already 4 days after treatment initiation, the average tumor volume from the treated groups was significantly reduced (6.1% and 9.7% for propranolol and ICI, respectively), with the antitumor effect becoming significantly stronger in the case of ICI on day 35 after treatment initiation (*p* = 0.0427). Propranolol treatment also resulted in a significant tumor growth reduction (as shown on day 50, *p* = 0.0123). At the end of the study, on day 58 (21 days after treatment initiation), both propranolol and ICI were able to significantly inhibit tumor growth (26% and *p* = 0.0141 and 20% and *p* = 0.0131, respectively), and were able to create a 7-day window of tumor growth retardation (at day 50). We also observed a reduction in tumor weight by 8% and 24% upon propranolol and ICI treatment (*p* = 0.2712 and *p* = 0.0961, respectively) (Figure 5C). Moreover, despite the daily treatment, no adverse effects were observed during the whole time of the treatment, followed up by animal behavior observation and weight evolution.

Finally, expression of HIF-2α protein on tumor samples was addressed immunohistochemically. Figure 5D shows a clear reduction of HIF-2α expression on ADRB blockers-treated mice, as observed in vitro (Figure 2A).

These in vivo data demonstrate the antitumor properties of propranolol and ICI and the lack of major side effects when tested in a murine model.

### 2.7. Data from VHL Patients Treated with Propranolol as Off-Label Prescription

VHL patients bearing retinal-HB tumors were prospectively treated with propranolol as off-label prescription, following the related literature [19,20] and the orphan drug designation of propranolol for VHL. Table 2 shows the RCC status, up to the present, from 4 VHL patients affected with ccRCC, retinal-HBs, and CNS-HBs. In addition, kidney MRI from the 4 different patients are presented in Figure 6, showing the lesions status when patients started with propranolol treatment, and the follow-up in time, as indicated by the dates. Propranolol-treated patients were followed according to the VHL clinical protocols. All the detected renal lesions were measured. Though we focus on those more significant lesions based on the characteristics and size to assess the treatment requirements. We also considered those cystic lesions feasible to measure in the follow-up. Considering the pathophysiology could be similar to solid tumors, we also observed the propranolol effects in the cystic lesion’s growth. Tumor growth was assessed by urologists and radiologists specialized in VHL, through enhanced MRI and/or ultrasound every six months. The treatment prescription, dose adjustment, and clinical monitoring were carried out by an internist with extensive experience in VHL patient management. During the observation time, all patients remained stable, with neither progression of the preexisting ccRCCs nor development of new carcinomas.

Patient 1: Since she started propranolol treatment 36 months ago, she has not developed any kind of renal lesions along the follow-up. Her mother, also a VHL patient, underwent a total bilateral nephrectomy due to multiple kidney tumors (Figure 6A).

Patient 2: He started propranolol treatment after a partial nephrectomy due to a solid and cystic lesion which resulted in ccRCC and more small bilateral cystic lesions were detected. After 22 months of follow-up, no changes in these cyst lesions have been observed (Figure 6B).

Patient 3: He underwent a bilateral partial nephrectomy before propranolol treatment started. After six months of treatment, a recurrent solid tumor grew from 23.4 to 30.0 mm in his right kidney which was resected in September 2017. No relapsing or new solid tumors have been detected. Along the 42 months with propranolol treatment, two main cystic lesions in his left kidney were detected, and grew from 12.5 mm to 15.9 mm, and from 9.0 mm to 23.2 mm (Cyst 1 and 2, respectively). None of them exhibited solid component or surgical criteria for VHL at the last control. The rest of the renal parenchyma show multiple small bilateral cystic lesions, remaining stable in the same period (Figure 6C).

Patient 4: 8 months before propranolol treatment initiation, three cystic lesions of 4, 6, and 9 mm in diameter each were detected. These lesions remain without changes during 16 months of follow-up under propranolol treatment (Figure 6D).

The main constraint of this study is the limited size of the cohort (comprising 4 VHL patients). However, the data obtained are consistent with the in vitro and in vivo results. These data provide further biological support for the clinical application of β2-blockers and their beneficial role in the treatment of VHL-related neoplasms. Moreover, the lack of side effects after a systemic and chronic treatment (follow-up ranges from 15 to 47 months) corroborates the safety of use of this drug on VHL patients. Finally, the protective effect of propranolol and other β-blockers has been reported in other retrospective population-based studies examining hepatocellular carcinoma, prostate cancer, and other malignancies [43].

## 3. Discussion

In VHL patients, a second stochastic event (e.g., missense/nonsense mutation of the *VHL* gene) ends in a defect or absence of pVHL. Hence, the lack of functional pVHL in VHL patients triggers the initiation of multiple highly vascularized tumors alongside their life span (resulting in an average life expectancy of 59.4 years for males and 48.4 years for females) [44]. In addition, VHL-related mortality is mainly due to complications derived from HB-CNS and ccRCC tumors [44].

Currently, repeated surgeries are the only therapeutic option for VHL patients. Nevertheless, surgeries do not resolve the preoperative neurological deficits since there is a cumulative morbidity in neurologic and retinal functions after each consecutive surgery [45]. After HB-CNS surgical resection, only 20% of the patients maintain or improve their preoperative symptoms, generating debilitating conditions [46,47]. In addition, partial nephrectomy is the only alternative to a complete kidney resection in the case of ccRCCs. Therefore, the lack of therapies for diffuse or recurrent symptoms leads to an urgent demand of effective drugs for VHL patients, especially those that might halt the tumor progression and, subsequently, delay surgical interventions.

To our knowledge, 19 different VHL-related interventional clinical trials have been registered by the U.S. National Library of Medicine and the EU Clinical Trials Register (Table 1). Half of them are exclusively focused on RCC and two-thirds are based on conventional chemotherapeutic drugs, which are able to target rapidly proliferating cells in both normal and cancer tissues: 10 were focused on the role of TKI, mainly sunitinib, sorafenib and pazopanib, and another 5 aimed at blocking VEGF with bevacizumab or its Fab fragment ranibizumab (Table 1).

Most of the systemic therapies assayed have shown limited responses in VHL pancreatic and renal tumors [14,15], while CNS HB had a very low response. Moreover, they are often associated with severe toxicities and side effects, which were often the reason for their discontinuation and termination [14,15].

An appealing alternative would be to identify new therapeutic agents that target vulnerabilities associated with the genetic and epigenetic properties of specific tumor types. In this regard, few of the recent trials target HIF-2α (PT2799), Hsp90 (17AAG), or β-adrenergic receptor (propranolol) are aiming to open new strategies for VHL treatment (Table 1). Among them, the selective, small molecule HIF-2α inhibitor PT2799 has recently reported its results on 61 VHL patients bearing an early stage nonmetastatic ccRCC [34]. The data show a limited response since only 27.9 showed partial response and none showed a complete response; nevertheless, some responses were observed on CNS, retinal, and pancreatic lesions without severe adverse effects as was the case with TKI such as pazopanib (Table 1) [14,15]. The authors point to a phase 3, probably changing their dosage and duration of the trial. This study shows that different strategies than TKI or VEGF-blockers are demanded for VHL.

According to the literature, many different tumors are driven by ADRB receptors [48,49,50]. Moreover, some papers have shown in vitro the therapeutic benefits of ADRB-2 blockers, counteracting the activation of the receptor [51,52,53]. Previous works on infantile hemangioma (IH) [54,55,56,57,58] and CNS-HB primary tumors [16,18] demonstrated the in vitro antitumoral effects of propranolol, an ADRB-1,2 blocker. Propranolol, via the β2-receptor, triggered apoptosis and decreased HIF levels and its nuclear localization on CNS-HBs *VHL^−/−^* primary tumors, as well as the specific ADRB-2 antagonist ICI. Moreover, in a phase III clinical trial conducted in 7 VHL patients harboring retinal HB, tumor growth was halted for the duration of the treatment with propranolol [19,20]. These results led to the designation of propranolol as an orphan drug for VHL in 2017 by the European Medicine Agency (EU/3/17/1841).

ccRCC is the first cause of death in VHL and comprises 63% of the RCCs in the general population. Both RCCs share a unique and common characteristic: a homozygous mutation in the *VHL* gene, which drives the stabilization of HIF protein in the resulting tumors, with HIF-2α being the predominant isoform. [36]. Hence, targeting HIF might have beneficial antitumor effects in both VHL and RCC patients.

Despite the remarkable differences between VHL-HBs and VHL-ccRCCs, such as the type of cells (endothelial vs. epithelial), the common pseudohypoxia state due to loss of functional pVHL makes HBs and ccRCCs suitable for finding common therapeutic strategies. Thus, the purpose of this study was to address whether β2-adrenergic antagonists, such as propranolol and ICI, were also effective in ccRCC, as they are in HBs [18], and if the mechanism of action could be similar too.

To the best of our knowledge, this is the first study with primary RCC culture-derived tumors resected from VHL patients (Figure 1A). Additionally, we used the *VHL*^−/−^ 786-O human ccRCC cell line, which represents a useful model to study VHL-related tumors. Using this compendium of line and primary ccRCC cell cultures, we explored the antitumor potential of the two β2-blockers, propranolol and ICI.

As in HBs [18], propranolol and ICI showed similar capacity to impair cell proliferation in RCC cells, in a dose-dependent manner, with a LD50 higher than in the 786-O cell line (Figure 1B, Appendix A). Figure 1E,F show that apoptosis is the main cause of decreased viability. Two pro-apoptotic genes, *BAX* and *CASP9*, were upregulated after treatment with propranolol and ICI, in agreement with the measurements of caspase 3/7 activity and to the microscopy images of nuclear integrity.

HIFs are probably the most favored molecules in *VHL^−/−^* cells; free from degradation, they can accumulate and translocate into the nucleus, triggering the expression of a large number of genes [4]. HIF-2α protein expression was addressed by Western blot in untreated cells or after either 100 µM propranolol or ICI treatment. In vitro HIF inhibition was apparently stronger in the 786-O cell line than in the primary tumors after a β2-blocker treatment, although a complete effect on a decrease in the HIF-2α protein levels could not be achieved (Figure 2A). In this context, it is interesting to mention that physapubescin (from *Physariumpubescens*) has been shown to react with protein thiol-nucleophiles, down-regulating the expression of HIFs and selectively up-regulating the expression of *CHOP* and *DR5* leading to apoptosis in VHL-null RCC [59].

Secretion of VEGF, a direct HIF target, was reduced both in 786-O cells and in VHL-ccRCC primary tumors after β-antagonists treatment (Figure 2B). Furthermore, the RCC biomarker and direct target of HIF, CAIX, is also downregulated (and in a similar way as with HIF-2α) after propranolol or ICI treatment. This result correlates with our previous findings in HBs [16,18] indicating that the mechanism of action in RCCs is similar to that observed in HBs, and that in both cases we are interfering with a common link: the constitutively active HIF proteins, interfered by the β2-adrenergic blockers.

Propranolol and ICI also exerted gene expression changes in different VHL-ccRCCs, such as the already mentioned pro-apoptotic genes *BAX* and *CASP9* and the HIF target gene *AQP-1*. *AQP-1* is of special interest in VHL-HBs since *AQP-1* encodes a transmembrane water channel protein [37,38,39]. *AQP-1* expression is induced by HIFs and its enhanced expression may increase liquid flow across the cell membrane, leading to cystic growth, commonly described in VHL tumors. The observation of a significantly decreased *AQP-1* expression, similar to the observed in HBs [18], would lead to slowing down of the cystic growth surrounding the tumor. In addition, *ADRB-2* expression itself was downregulated by both propranolol and ICI (Figure 2C and Appendix A), indicating that ccRCCs can be targetable tumors by *ADRB-2* antagonists. It is worth noting that *ADRB-1* is not expressed by the ccRCCs, according to our PCRs results (Appendix A).

Considering that ADRB-2 antagonists reduced HIF levels, and significantly inhibited HIF target genes (such as *VEGF*, *AQP-1*, and *CAIX*), HIF-2α subcellular distribution had to be addressed. We had previously reported that HIF-1α nuclear internalization on CNS-HBs and primary HUVECs under chemical hypoxia was impaired after treatment with β2-blockers [18]. Here, we confirm that after propranolol or ICI treatment, HIF-2α nuclear translocation was similarly impaired (by 24% and 21%, respectively) (Figure 3). Although we cannot formally confirm it, it seems reasonable to speculate that changing the subcellular localization of HIF-2α and thus their transcriptional activity, could lead to a clinical improvement in RCC tumor patients [60]. Unlike HIF-1α, HIF-2α also plays a major role in the tumor cytoplasm. Thus, Uniacke et al. demonstrated that HIF-2α can form a complex in the cytoplasm to help the initiation of protein synthesis in periods of oxygen scarcity [57,61]. This mechanism was shown to be especially important for tumor development as it allows the tumor cells to overcome the hypoxia-induced repression of protein translation, which can explain the correlation between high HIF-2α cytoplasmic abundance and unfavorable prognosis in RCC patients. Although the data here shown throws light on the presence and distribution of HIF-2α on RCCs, the exact molecular mechanism by which HIF-2α is contributing to VHL-RCC malignancy remains unresolved.

On the other hand, hypoxia or a pseudo-hypoxic status caused by absence of *VHL* expression, induces the activity of the *NFκB* pathway—another major molecular pathway triggered to mediate cellular responses upon hypoxic stimuli. *NFκB* induced by hypoxia can independently upregulate many inflammatory genes and directly induce *HIFs* gene expression [58,62]. Although the mechanism underlying *NFκB* activation by hypoxia remains elusive, one well-accepted mechanism is that, under hypoxic conditions, inhibition of *NFκB* is activated by IKK-mediation [59,63]. In any case, the combination of both programs—the *HIF* genes and the inflammatory antiapoptotic program triggered by *NFκB* activation—may contribute to tumor malignancy and drug resistance. Therefore, the ideal anti-tumoral drug should tackle both HIFs and NFkB pathways. In the study here presented, by means of immunofluorescence microscopy, propranolol and ICI show a significant reduction of the nuclear translocation of p65 (32% and 43%, respectively) (Figure 4A). Altogether, ADRB-2 antagonists have proven to be able to reduce the translocation of HIF-2α and p65, master transcription regulatory keys of both pathways. IL-1β, IL-6, and TNFAIP6 are known targets of p65, the inflammatory program led by NFkB [58,62]. Propranolol or ICI treatments significantly decreased the expression of these p65 target-genes as observed by q-PCR (Figure 4B), which confirms the highly promising therapeutic properties of the tested ADRB-2 antagonists.

Results of our two in vivo xenograft experiments with 786-O cells on NSG mice suggest that propranolol and ICI slow down tumor progression in two different approaches: (i) early short-time treatment (“prevention setting”) (Figure 5A) and (ii) treatment of an established tumor (“therapeutic setting”) (Figure 5B). Propranolol and ICI delayed tumor growth by up to seven and ten days, creating a therapeutic time window with applications in a clinical scenario for further surgeries or combinatorial therapies. The delay in the tumor establishment and tumor growth in vivo may be explained by a decrease in serum VEGF-A levels. This decrease may be of human origin (from the 786-O cells) and of murine origin (induced by the tumor cells in surrounding stroma). In this way, propranolol and ICI may have a direct anti-tumor effect by inducing apoptosis. This fact has been supported by the observed induction of *BAX-2* and *CASPs 3/7/9*, as also previously shown in HBs [16,18]. Additionally, β2-blockers may also induce systemic changes that lead to a poorly adequate tumor microenvironment. On one hand, β-adrenergic antagonists are acting as pro-apoptotic, and on the other hand, they act as anti-angiogenic agents by reducing HIF activity and therefore leading to a downregulation of HIF targets like *CAIX* and *VEGF*. Plasma VEGF levels were recently reported to serve as a reliable biomarker in patients with VHL taking propranolol for retinal HBs [19,20]. While tumors remained stable, VEGF plasma levels in VHL patients decreased to standard levels during the treatment, in agreement with our in vitro and in vivo findings.

In addition, our prospective analysis of ccRCC growth kinetics in VHL patients with retinal tumors receiving propranolol treatment in an off-label use, following the results of Albiñana et al. [19], revealed a stable status with cessation of growth of RCCs during the whole period of propranolol treatment (Table 2). Considering the cardiovascular symptoms of propranolol, its dose escalation was based on the tolerance of hypotension or bradycardia. An increase of 20–40 mg in the dose of propranolol happened when the cardiac rate was higher than 60 per minute and the arterial pressure was equal to or higher than 100/60 mmHg in absence of symptoms. Moreover, presence of other symptoms like fatigue or sleep deprivation nightmares was studied.

The measurements were made by VHL-trained radiologists belonging to the VHL Unit at Hospital Italiano de Buenos Aires (Argentina). The patients had been imaged by 320 CTMS or 3.0 Tesla Ingenia Siemens MRI (Siemens, Erlangen, Germany) and the findings were discussed in multidisciplinary clinical meetings in which a detailed comparison with previous tests was carried out (Figure 6).

For this prospective study, we have considered “stable” cases when the lesion growth is not significant in a period between 6–12 months, especially when the size of the lesions does not require a surgical treatment. However, we cannot know about the natural history of each lesion in these patients without propranolol. Despite this, it seems that the lesions tend to have a slower growth, especially in patient 1, sharing a mutation that provoked a total bilateral nephrectomy for her mother but no tumor growth for her, and in patient 3 who has a history of several and fast growth lesions in the past.

Despite the limited group size and the lack of a priori strategy, these data are not conclusive but consistent with our cell culture and in vivo studies and thus provide further rationale for using propranolol in the treatment of VHL-related neoplasms. The follow-up of the presented 4 cases and the search for new data from more VHL patients is urgent, and the commitment will be made. Moreover, the protective effect of propranolol, previously described by us for CNS-HB [16,18], was supported by Shepard et al. in 2018 [36]. Lastly, other β-blockers have been reported in other retrospective population-based studies examining hepatocellular carcinoma, prostate cancer, and other malignancies [43].

Propranolol shares affinity for ADRB-1 and -2 and therefore comes with the disadvantage of cardio-specific side effects. Hence, a highly specific ADRB-2 antagonist may be the ideal compound to treat VHL patients, as it would maintain the therapeutic properties of propranolol while avoiding its hypotension and bradycardia effects. Thus, since we have shown clear in vitro differences in LD50 between non-cancerous cells and ccRCC cultures, and no undesirable side effects have been observed in either animal models or clinical trials [18], we believe it would be beneficial to initiate ICI for clinical use and even try applying escalating doses.

The blockade of the ADRB-2 pathway opens a new strategic avenue for the anti-tumor treatment spectrum. Overall, TKIs have shown limited response in VHL patients, particularly in ccRCC, where a 42% partial response has only been reported in one of the latest trials [15]. Since ADRB-2 blockers (similar to TKIs) primarily act as antiangiogenics, a combination of TKIs and ADRB-2 blockers might have synergic effects, improving the results of individual treatments. On the other hand, in aggressive and metastatic tumors, a combination of intercalating agents targeting cellular division with ADRB-2 blockers might reduce metastatic dissemination, as has been recently shown in metastatic paraganglioma [60,64].

Remarkably, ADRB-2 blockers target receptors on the surface of the cells and, in contrast to TKIs and intercalating agents, their internalization is not required. This is an added advantage since this will prevent from the development of multidrug resistance mechanisms (MDR) such as by ABC transporters, controlling drug exclusion from the cancer cells. These mechanisms should not be operative with ADRB-2 blockers. In this respect, preliminary results obtained by our group have shown a limited resistance of 786-O cells to propranolol. Gene expression of the few resistant cells was analyzed by RT-qPCR, supporting a decrease in expression of *MDR* genes (Appendix A). Thus, the only expected mechanism of resistance to ADRB-2 blockade could be a decrease in the expression levels of ADRB-2 receptors, as it is demonstrated in Appendix A. However, as reported, many tumors are driven through ADRB-2 activation [48,49,50,61,65], and a decrease in the receptor expression would lead, in turn, to a decrease in tumorigenicity [51,52,53].

In summary, these findings demonstrate for the first time that ADRB-2 antagonists could be used as promising therapeutic agents to treat RCCs, acting through inhibition of cancer cell proliferation, tumor angiogenesis, and inflammation (Figure 7). These findings support further implementation of β-blockers into the clinic as a promising new treatment for VHL and other non-VHL tumors that share molecular similarities with VHL.

## 4. Materials and Methods

### 4.1. Ethics Approval and Consent to Participate

All methods were carried out in accordance with Consejo Superior de Investigaciones Científicas CSIC and Spanish guidelines and regulations. All experimental protocols were approved by the CSIC Ethical Committee (code 075/2017), including human samples handling, and were done following the guidelines of the World Medical Assembly (Declaration of Helsinki). Animal experiments were approved by both CSIC committee of animal welfare and Madrid animal experimental committee (PROEX 045/17). Both institutions follow the “Ethical Principles and Guidelines for Experiments on Animals” from the Swiss Academy of Medical Sciences (SAMS) and the Swiss Academy of Sciences (SNAT).

Informed consent was obtained from all subjects whose surgical surplus were utilized to generate primary tumor cultures. All primary tumors derived from human samples are part of a clinical data collection registered at the Instituto de Salud Carlos III (ISCIII).

### 4.2. Cell Culture and Treatments

Clear cell renal cell carcinoma primary tumor cultures (ccRCC) were obtained from surplus of resected surgery tumor samples from VHL patients following the procedure previously described (16), and cultured in RPMI medium supplemented with 20% fetal bovine serum (FBS), 2 mM L-glutamine, and 100 U/mL penicillin/streptomycin (all from GIBCO, Grand Island, NY, USA). All patients provided written informed consent to use their tissue samples for this study.

HUVECs (ATCC-CRL-1730) were cultured in EGM-2 (Lonza, Walkersville, MD, USA) supplemented with 10% FBS, 2 mM L-glutamine, and 100 U/mL penicillin/streptomycin (GIBCO, Grand Island, NY, USA).

The human renal cancer cell line 786-O (ATCC CRL-1932) was cultured in RPMI supplemented with 20% FBS, 2 mM L-glutamine, and 100 U/mL penicillin/streptomycin.

ccRCC primary tumors, 786-O cells, and HUVECs were incubated with different doses of ADRB antagonists for the time and dose indicated in each experiment. Atenolol (Sigma-Aldrich, St. Louis, MO, USA) was dissolved in DMSO (Merck, Darmstadt, Germany), while propranolol and ICI (Sigma-Aldrich, St. Louis, MO, USA) were dissolved in distilled water.

All the cellular assays were performed at 37 °C, 5% CO_2_ and humidity conditions.

### 4.3. Cell Viability Assay

The viability of the ccRCC primary tumors, 786-O cells, and HUVECs was measured using the “Luminescent Cell Viability Assay” (Promega, Madison, WI, USA). This is a homogeneous quantitative method to determine the number of viable cells in culture based on quantitation of the ATP presence, which indicates metabolically active cells.

A total of 5000 cells/well were plated in 96-well plates and cultured in 100 μL with [0- 25- 50- 100- and 200 μM] propranolol or ICI for 72 h. Then, 100 μL/well of Cell Titer-Glo reagent (Lysis buffer, Ultra-Glo Recombinant Luciferase, Luciferine and Mg^2+^) was added and gently mixed for 15 min at RT. Finally, luminescence was measured using a Glomax Multidetection System (Promega, Madison, WI, USA) [16,18].

### 4.4. Caspase Activation Assay

The Caspase-Glo^®^ 3/7 Assay (Promega, Madison, WI, USA) is a luminescent assay that measures caspase-3 and caspase-7 activities by using a luminogenic caspase-3/7 substrate composed of the tetrapeptide sequence DEVD, luciferase, and cell lysis buffer. Therefore, the luminescence generated is proportional to the amount of active caspase presence.

A total of 5000 cells/well were plated in 96-well plates and cultured in 100 μL with [0- 25- 50- 100- and 200 μM] propranolol or ICI for 72 h. Then, 100 μL/well of Caspase Glo^®^ 3/7 Reagent (Promega, Madison, WI, USA) (Lysis buffer, Ultra-Glo Recombinant Luciferase, DEVD-aminoluciferine, and Mg^2+^) was added and gently mixed for 1 h at RT. Finally, luminescence was measured using a Glomax Multidetection System (Promega, Madison, WI, USA) [16,18].

### 4.5. Real-Time Quantitative PCR

Total RNA was extracted from cell cultures using Nucleo Spin RNA kit (Macherey-Nagel, Düren, Germany). One microgram of total RNA was reverse-transcribed in a final volume of 20 μL with High Capacity cDNA Reverse Transcription Kit supplemented with an RNase Inhibitor (Thermo Fisher Scientific, Vilnius, Lithuania) using random primers. FastStart Essential DNA Green Master (ROCHE, Mannheim, Germany) was used to carry out real-time PCR using an iQ5 system (BioRad, Foster City, CA, USA). Table 3 shows primers used for q-PCR.

### 4.6. Western Blot Analysis for HIF-2α

Primary ccRCC tumor cultured cells and the 786-O ccRCC cell line were incubated with 100 μM propranolol, ICI, or vehicle for 48 h. Then, cells were lysed on ice for 30 min in TNE buffer (50 mM Tris, 150 mM NaCl, 1 mM EDTA, and 0.5 % Triton X100 (Sigma-Aldrich, St. Louis, MO, USA) supplemented with wide-range protease inhibitors (Roche, Basel, Switzerland) and lactacystin (Sigma-Aldrich, St. Louis, MO, USA), a specific proteasome inhibitor to preserve HIF integrity. Lysates were centrifuged at 14,000 g for 5 min. Similar amounts of protein from cleared cell lysates were boiled in SDS sample buffer and analyzed by 4–20% SDS-PAGE under non-reducing conditions (BioRad, Foster City, CA, USA). Proteins from gels were electro-transferred to nitrocellulose membranes (Amersham, Little Chalfont, UK) followed by immunodetection with anti-HIF-2α (NOVUS, Oxon, UK) and anti-actin (Sigma-Aldrich, St. Louis, MO, USA) antibodies. Following primary antibody incubation, samples were washed and incubated with the corresponding horseradish peroxidase-conjugated secondary antibodies from Dako (Glostrup, Denmark). All antibodies were used at the dilution recommended by the manufacturer. Membranes were developed by chemiluminescence (SuperSignal West Pico Chemiluminescent Substrate, Thermo Scientific, Rockford, IL, USA).

### 4.7. VEGF Determination

Quantikine Human VEGF ELISA kit from R & D Systems (R & D, Abingdon, UK) was used to quantitatively determine human VEGF concentrations in supernatants from ccRCC primary tumors and 786-O cell line treated with vehicle or 100 μM propranolol or ICI for 48 h.

### 4.8. Immunofluorescence Microscopy

Immunofluorescence analyses were performed to evaluate the effect of propranolol and ICI on the subcellular distribution of HIF-2α and p65, and to examine the expression of CAIX, a cell marker of ccRCCs. Therefore, 5 × 10^3^ 786-O cells were seeded on sterile coverslips (13 mm diameter, VWR international, Radnor, PA, USA) placed at the bottom of a 24-well plate. On the next day, cells were treated with 100 µM either propranolol or ICI for 48 h.

Then, cells were washed with PBS and fixed with 3% paraformaldehyde (PFA) for 10 min at RT. After two PBS washing steps, samples were incubated with blocking solution (1% goat serum and 1% BSA in PBS) for 1 h at RT.

Cells were incubated overnight at 4 °C with mouse anti-human HIF-2α (1:100) (NOVUS), p65 (1:400) (Cell Signaling, Danvers, MA, USA), or CAIX (1:100) (Dako, Santa Clara, CA, USA). Following this, cells were washed thoroughly four times with PBS and incubated for 1 h at RT with goat anti-Mouse IgG (H+L)-Alexafluor 568-conjugated antibody (1:200) (Thermo Fisher Scientific, Waltham, MA, USA). Finally, cells were washed with PBS and coverslips were mounted on glass slides using Prolong+DAPI mounting media (Molecular Probes, Eugene, OR, USA). Using the fluorescence confocal microscope SP5 (DMI6000 CS Leica Microsystems, Wetzlar, Germany), 40× confocal images were taken. Green, red, and blue channels represent HIF-2α/p65, CAIX, and DAPI stains, respectively.

For the subcellular distribution determination, DAPI nuclear signal was used to fix the upper and lower sample limits (z-axis) and 1 µm z-stacks were programmed. After a maximum intensity projection, DAPI signal was used to identify and distinguish the nuclear limits and the HIF-2α or p65 signal to identify the cellular limits.

FIJI-ImageJ software (NIH, MD, USA) was used to measure the fluorescent intensities as follows. Using the HIF-2α or p65 signal, a straight line alongside the longer diameter and crossing the cell nuclei was traced and its intensity were measured (the whole cell signal). Then, using the nuclear limits, the very same line was shortened to reach just the nuclei and its fluorescence intensity was again measured (nuclear signal). Finally, the whole cell signal was normalized to 100% and, using the nuclear signal, the ratio nucleus/cytoplasm percentage was rated.

For CAIX expression levels, DAPI nuclear signal was used to fix the upper and lower sample limits (z-axis) and 1 µm z-stacks were programmed. After a maximum intensity projection, individual CAIX cell intensity was measured using FIJI-ImageJ software (NIH, MD, USA). Finally, the PBS group cell signal was normalized to 100% and the intensities from the cells treated either with propranolol or ICI rated.

### 4.9. In Vivo Tumor Xenografts

In order to address the antitumor properties of β-blockers, two different heterotopic in vivo approaches were employed. Both animal experiments were performed according to the approved in vivo experimental procedure PROEX 045/17.

In the first in vivo study, 10^6^ 786-O cells were subcutaneously injected in the dorsal flank of 7–8-week-old NOD scid gamma (NSG™) male mice (kind gift of Dr Garcia-Sanz JA, own breeding, originally from The Jackson laboratory, Bar Harbor, ME, USA). On the next day and during the first five consecutive days after implantation, two groups were treated intraperitoneally either with 10 mg/kg body propranolol, ICI-118,551, or Vehicle (9–10 mice per group). Tumor growth was measured twice by a caliper every 2–3 days and volumes were calculated according to the formula: shortest^2^ × largest × 0.52. Mice were sacrificed when the control group (Vehicle) reached an end point established according to ethical procedures.

In the second in vivo experiment, 10^6^ 786-O cells were subcutaneously injected in the dorsal flank of 7–8-week-old NSG™ male mice. Tumor growth measurements and calculations were assessed as mentioned above. When average tumor volume reached 100 mm^3^, the mice were randomly divided in 3 groups (9–10 mice per group) and each group was daily treated intraperitoneally with either 10 mg/Kg body propranolol, or ICI-118,551, or Vehicle for 20 more days. The mice were euthanized when the control group (Vehicle) reached an end point established according to ethical procedures.

### 4.10. Immunohistochemistry

HIF-2α expression from tumor xenografts was evaluated in paraffin-embedded tumor xenografts belonging to the “Classical in vivo cancer research assay”. The immunohistochemical staining was done in 5 μm deparaffined and hydrated sections. Sections were incubated with an anti-HIF-2α antibody (1:600) (NOVUS, Oxon, UK). Then, samples were washed and incubated with HRP-conjugated secondary mouse anti-human antibody. HRP activity was amplified with DakoEn Vision + Dual Link System-HRP (Dako, Santa Clara, CA, USA) and the visualization was performed with a DAB substrate Kit (Dako, Santa Clara, CA, USA). Samples were counterstained with hematoxylin 0.02% and mounted with DPX mounting medium (Sigma-Aldrich, St. Louis, MO, USA). Images were taken at 10× with an Olympus digital camera (Olympus, Hamburg, Germany) coupled with an Axio Vert A1 Zeiss microscope (Zeiss, Jena, Germany). The Vehicle group was tested as positive control and negative controls were performed in parallel, with omission of the primary antibody incubation step.

### 4.11. Clinical Data from Patients

We conducted a retrospective analysis of patients who were treated with propranolol because of a retinal HB but also bearing a ccRCC. Their medical histories were analyzed in order to determine whether the treatment with propranolol influenced the renal tumor development and progression. Hence, parameters like propranolol dosage, treatment duration, tumor growth, recurrence after resection and metastases diagnosis have been collected and studied.

### 4.12. Statistical Analysis

Results are presented as mean ± SEM. Statistical analyses were performed using Student’s *t*-test. Statistical significance was defined when *p* < 0.05 (* *p* < 0.05; ** *p* < 0.01, and *** *p* < 0.001).

## 5. Conclusions

VHL-ccRCC, the second cause of death in VHL patients, lacks an effective systemic treatment, with recurrent surgeries being the only effective solution. The ADRB-2 antagonists propranolol and ICI-118,551 have shown therapeutic benefits on ccRCC in vitro and in vivo—triggering key processes like apoptosis, targeting and impairing the expression and signaling of key molecules such as HIF and NFĸB/p65, and showing anti-tumoral effects in vivo. In addition, clinical data show a better outcome of VHL-ccRCC patients after propranolol off-label treatment.

Propranolol and ICI-118,551 have shown antitumor benefits in VHL-derived ccRCC, and since ccRCCs comprise 75% of the total RCCs, targeting ADRB-2 becomes a promising drug for VHL and other non-VHL tumors.

## Figures and Tables

**Figure 1 jcm-09-02740-f001:**
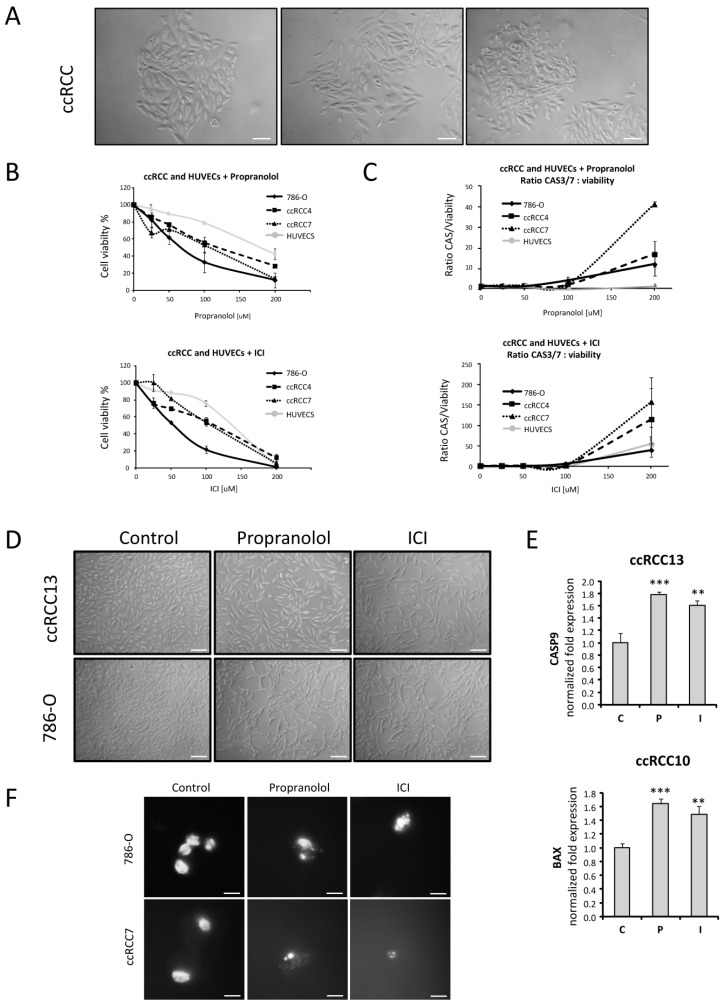
Effect of β2-adrenergic receptor (ADRB) blockers on cell viability of *VHL^−/−^* and *VHL^+/+^* cells. (**A**). Representative images of a primary clear cell renal cell carcinoma (ccRCC) tumor culture derived from a VHL patient. (**B**,**C**). Effect of β1 and β2-adrenergic receptor blockage in cell viability and Caspases 3/7 activation after propranolol and ICI-118,551 (ICI) incubation on *VHL^−/−^* ccRCC cells such as 786-O, ccRCC4, and ccRCC7 and on the *VHL^+/+^* endothelial primary cells HUVECs. Cultures were treated with increasing doses [0–200 µM] propranolol or ICI for 72 h. (**D**). Representative images of ccRCC13 and 786-O cultures treated with 100 µM propranolol or ICI for 72 h. (**E**). mRNA quantification of the apoptotic genes *CASP9* and *BAX* on ccRCC primary tumors treated with 100 µM propranolol or ICI for 72 h. (**F**). Propranolol and ICI cause chromatin condensation induced by cellular apoptosis. All data are based on 3 independent experiments. Scale bars represent 100 µm (**A**,**D**) and 50 µm (**F**). Abbreviations: C, P, and I for control, propranolol, and ICI, respectively. Error bars denote ± SEM. Student’s *t*-test: * *p* < 0.05; ** *p* < 0.01; *** *p* < 0.001.

**Figure 2 jcm-09-02740-f002:**
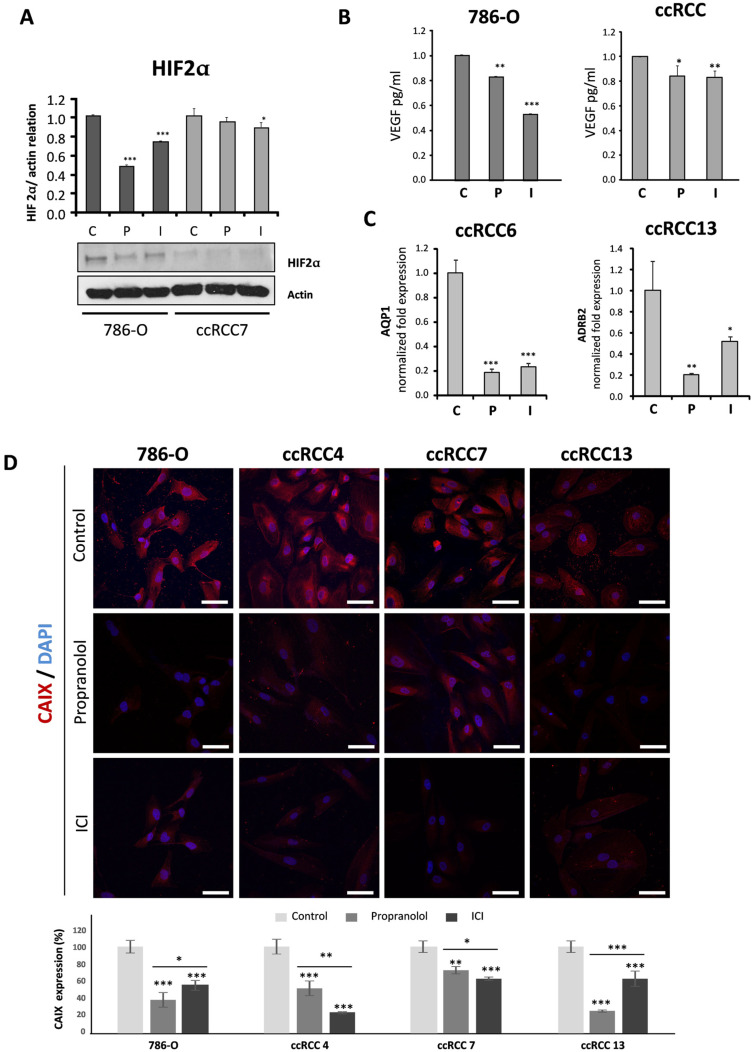
Effects of ADRB antagonists on VHL gene targets. (**A**). HIF-2α protein expression on 786-O and ccRCC cells after treatment with 100 µM propranolol or ICI for 72 h. Protein cell lysates were analyzed by Western blot, actin was used as loading control and for normalized quantification (representative assay). (**B**). Secreted VEGF levels in supernatants of cells treated with 100 µM propranolol or ICI for 48 h were measured by ELISA. VEGF data from ccRCC represents the average of three different tumor samples. (**C**). mRNA quantification of the HIF target genes *AQP-1* on ccRCC primary tumors treated with 100 µM propranolol or ICI for 72 h. (**D**). Immunofluorescence detection and confocal microscopy representative images from 786-O and ccRCC cultures treated with 100 µM PBS, propranolol, or ICI for 48 h. Up, mouse anti-human CAIX antibody stains (red) and DAPI nuclear staining (blue) merged images. Down, relative quantification of CAIX expression on each cell line after treatment. For quantification procedures, 20–50 cells were measured for CAIX intensity from 3 replicates per condition. Scale bars represent 50 µm. All data are based on 3 independent experiments. Abbreviations: C, P, and I for control, propranolol, and ICI, respectively. Error bars denote ± SEM. Student’s *t*-test: * *p* < 0.05; ** *p* < 0.01; *** *p* < 0.001.

**Figure 3 jcm-09-02740-f003:**
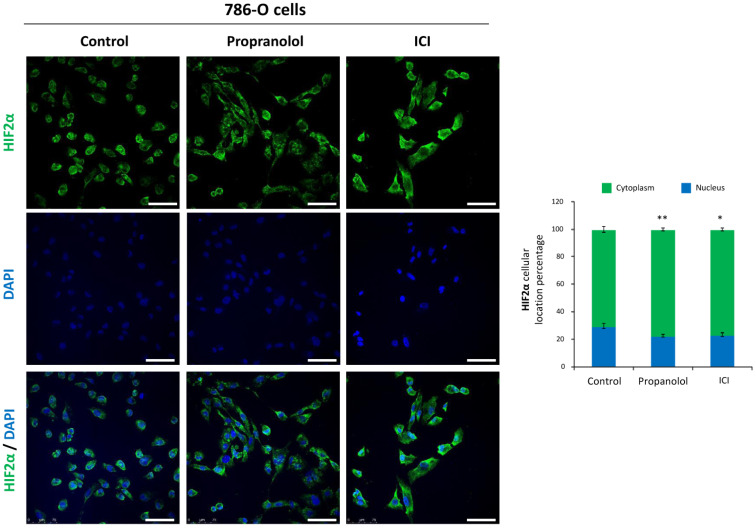
Effect of β2-adrenergic receptor blockage on HIF-2α subcellular distribution. Left, immunofluorescence detection and confocal microscopy representative images from 786-O cells treated with 100 µM propranolol or ICI for 72 h. Mouse anti-human HIF-2α antibody stains (green), DAPI nuclear staining (blue) and merged are shown. Right, relative quantification of HIF-2α nuclei or cytoplasmic distribution. For quantification procedures, 12 different optic fields were taken from 4 replicates per condition. Scale bars represent 50 µm. Error bars denote ± SEM. Student’s *t*-test: * *p* < 0.05; ** *p* < 0.01; *** *p* < 0.001.

**Figure 4 jcm-09-02740-f004:**
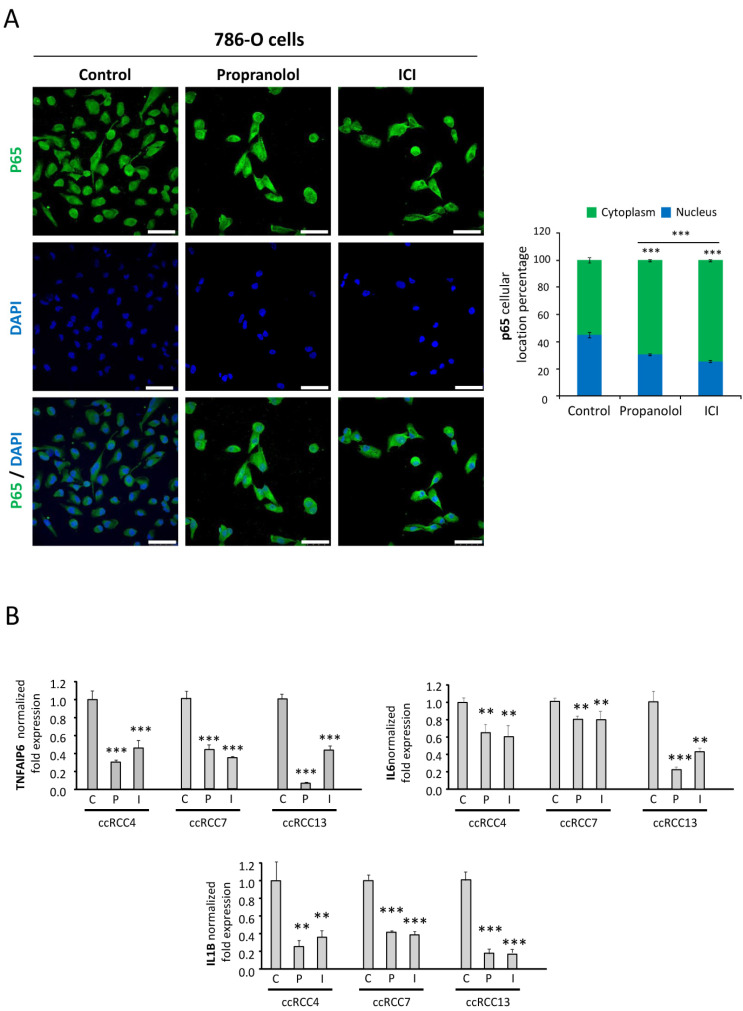
Effect of β2-adrenergic receptor blockage on p65 subcellular distribution. (**A**). Left, immunofluorescence detection and confocal microscopy representative images from 786-O cells treated with 100 µM propranolol or ICI for 72 h. Mouse anti-human p65 antibody stains (green), DAPI nuclear staining (blue) and merged are shown. Right, relative quantification of p65 nuclei or cytoplasmic distribution. For quantification procedures, 12 different optic fields were taken from 4 replicates per condition. Scale bars represent 50 µm. (**B**). mRNA quantification of the NFKB target genes: TNFAIP6 (up-left), IL-6 (up-right), and IL-1β (down). mRNA expression levels were measured in different ccRCC primary tumors after treatment with 100 µM propranolol or ICI for 72 h. Abbreviations: C, P, and I for control, propranolol, and ICI, respectively. Error bars denote ± SEM. Student’s *t*-test: * *p* < 0.05; ** *p* < 0.01; *** *p* < 0.001.

**Figure 5 jcm-09-02740-f005:**
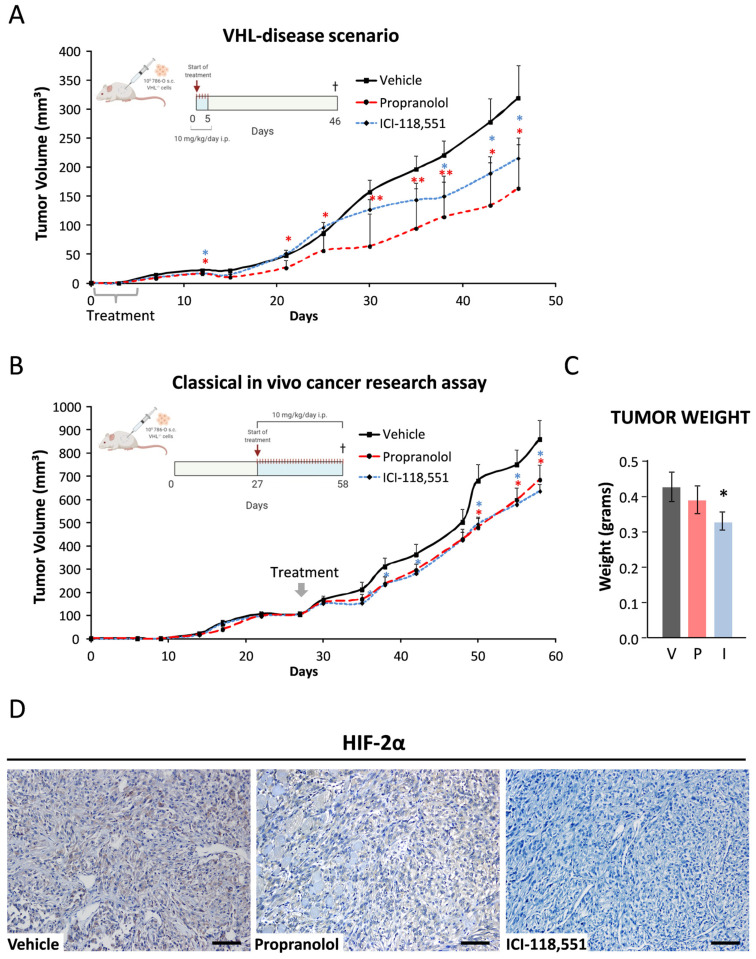
Inhibition of tumor growth by ADRB antagonists. Tumor volumes reached after subcutaneous injections of 10^6^ 786-O wild type cells (n = 9–10) and intraperitoneal treatment with 10 mg/Kg/day Vehicle (■), propranolol (●), or ICI (♦), in 10–12-week-old male NOD/SCID mice. (**A**). (VHL-disease scenario) Systemic treatment started along the first five days after tumor implantation; or (**B**). (Classical in vivo cancer research assay) when average tumor reached 100 mm^3^ volume. (**C**). Average weights of the collected tumors. (**D**). Immunohistochemical detection of HIF-2α in the ccRCC 786-O tumor xenografts. Cell nuclei were counterstained with hematoxylin. Scale bars represent 100 µm. (**C**,**D**) belong to the “(**B**). Classical in vivo cancer research assay” approach. Abbreviations: V, P, and I for vehicle, propranolol, and ICI, respectively. Error bars denote ± SEM for each time. Student’s *t*-test: * *p* < 0.05; ** *p* <0.01; *** *p* < 0.001.

**Figure 6 jcm-09-02740-f006:**
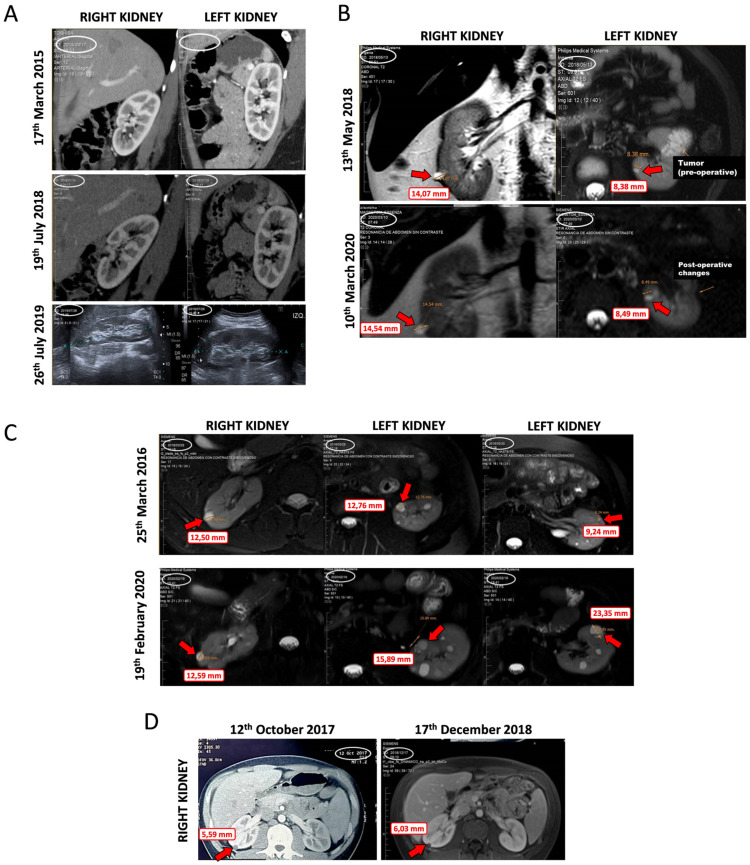
Different images of tumor and cystic lesions evolution in kidneys of VHL patients, before and during propranolol treatment. (**A**). Patient 1. Female who, after 36 months of treatment, shows no lesions. (**B**). Patient 2. Male, with a partial nephrectomy due to a solid and cystic lesion diagnosed as ccRCC and small bilateral cystic lesions detected. After 22 months of follow-up, the sizes of the cysts are similar without significant changes. No new tumor has appeared. (**C**). Patient 3. Male who underwent a bilateral partial nephrectomy before treatment and a tumor growth in his right kidney. No relapsing or new solid tumors have been detected since the treatment. Moreover, his left kidney showed two main cystic lesions, and none of them show surgical criteria, and multiple small bilateral cystic lesions remain. (**D**). Patient 4. Male, bearing three cystic lesions before propranolol treatment. These lesions remain stable after propranolol treatment.

**Figure 7 jcm-09-02740-f007:**
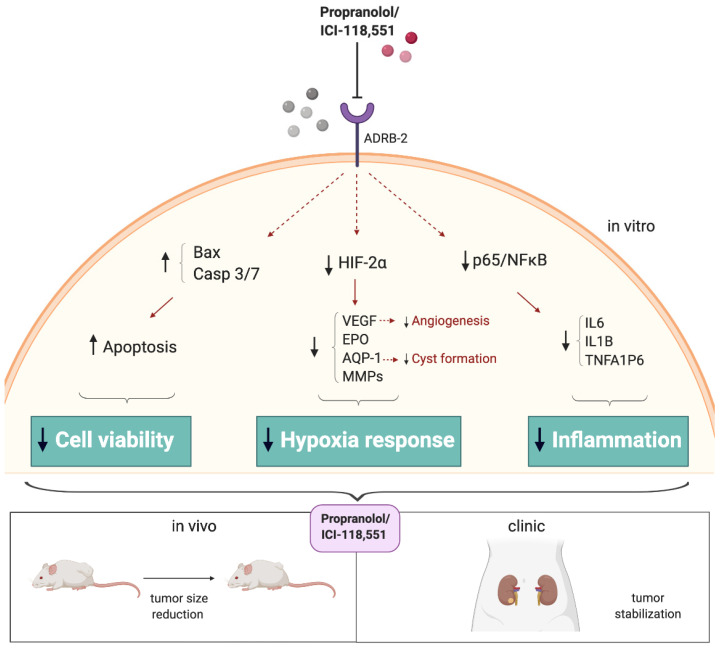
Antitumor benefits shown by ADRB-2 antagonists in VHL-derived ccRCC. Propranolol and ICI-118,551 have shown therapeutic benefits both in vitro and in vivo—on ccRCCs, by triggering key processes like apoptosis, increasing Bax or Caspases 3/7 expression, targeting and impairing the expression and signaling of molecules such as HIF and its targets, and decreasing NFĸB/p65 signaling, showing anti-tumoral effects in vivo. ADRB-2 antagonists could be used as promising therapeutic agents to treat RCCs, acting through inhibition of cancer cell proliferation, tumor angiogenesis, and inflammation.

**Table 1 jcm-09-02740-t001:** Compilation of the VHL interventional clinical trials registered at the EU Clinical Trials Register (EudraCT) (https://www.clinicaltrialsregister.eu) and the U.S. National Library of Medicine (NCT) (https://clinicaltrials.gov). Only interventional trials whether a therapeutic drug was tested are listed. All the trials showed here were designed as single-arm, open-label trials. Abbreviations: Status: C (Completed); R (Recruiting); O (Ongoing); T (Terminated).

Trial Registration	Phase	Title	Intervention	Number of Patients	Status Start Date	Outcome	Link to Results
NCT00052013	2	Treatment of Von Hippel-Lindau (VHL)-Related Hemangioblastoma With PTK787/ZK 222584	PTK787 ZK 222584 (vatalanib)	11	C Feb 2003	100% discontinued (adverse events).	[21]
NCT00056199	1	EYE001 to Treat Retinal Tumors in Patients With Von Hippel-Lindau Syndrome	EYE001(anti-VEGF aptamer)	5	C Mar 2003	80% showed stabilization or improvement of vision.	[22]
NCT00088374	2	17AAG to Treat Kidney Tumors in Von Hippel-Lindau Disease	1 DMAG	9	C Jul 2004	Study did not meet accrual.	[23]
			18FDG				
			[15-O] H2O				
			EPL diluent				
NCT00089765	1	Ranibizumab Injections to Treat Retinal Tumors in Patients With Von Hippel-Lindau Syndrome	Ranibizumab	5	C Aug 2004	Not clear therapeutic effect. “Minimal beneficial effects on most VHL-related Retinal Capillary HB”	[24]
NCT00330564	2	Evaluation of Sunitinib Malate in Patients With Von Hippel-Lindau Syndrome (VHL) Who Have VHL Lesions to Follow	SU011248 (Sunitinib)	15	T May 2006	Early termination due to slow accrual. 33% showed partial response in RCC but not in HB.	[25,26]
NCT00470977	1/2	Treatment of Exudative and Vasogenic Chorioretinal Diseases Including Variants of AMD and Other CNV Related Maculopathy	Ranibizumab injection (0.5 mg)	18	C May 2007	-	-
EudraCT 2007-002132-29	2	A Phase II Trial of Sorafenib (a tyrosine kinase inhibitor) given orally twice daily in renal cancer patients with vHL syndrome	Sorafenib	25	T Jan 2008	-	-
NCT00566995	2	Phase II Study of Vandetanib in Individuals With Kidney Cancer	ZACTIMA (Vandetanib) (ZD6474)	37	C Feb 2008	80% completed the study. None of them showed a complete response and 8% showed a partial response.	[27]
NCT00673816	1/2	Sunitinib Malate to Treat Advanced Eye Disease in Patients With Von Hippel-Lindau Syndrome	Sunitinib Malate	2	T May 2008	Study did not meet accrual plus adverse events.	[28,29]
NCT01015300	1	Bevacizumab (Avastin) in Unresectable/Recurrent Hemangioblastoma From VonHippel-Lindau Disease	Avastin	1	T Dec 2009	Study did not meet accrual.	-
NCT01168440 EudraCT 2009-013052-76	2	A single-arm, phase II study of SU11248 (sunitinib) in patients with von Hippel-Lindau (VHL) disease	Sunitinib	5	C Feb 2010	Disease progression (20%), unacceptable toxicity (60%), and lack of clinical improvement. after 7 cycles	[30]
EudraCT 2005-004502-82	2	A Phase II Study of Neoadjuvant Sunitinib in Metastatic Renal Cell Carcinoma	Sunitinib	16	T Oct 2010	Study did not meet accrual. 58% showed complete or partial response.	[31]
NCT01436227	2	Pazopanib in Von Hippel-Lindau (VHL) Syndrome	Pazopanib	32	O Jan 2012	80% discontinued (progressive disease, loss of quality of life, or intolerance). 42% showed partial response and 55% a complete or partial response in ccRCC.	[15]
NCT01266070	2	TKI 258 in Von Hippel-Lindau Syndrome (VHL)	Dovitinib	6	T Nov 2012	33% discontinued plus the trial met toxicity stopping rule.	[32,33]
EudraCT 2014-003671-30	3	Therapeutic effect of propranolol in a series of patients with von Hippel-Lindau disease and retinal hemangioblastomas in short, medium and long term treatment.	Propranolol	7	C Oct 2014	28% showed partial response and 72% a stable disease. No serious adverse effects were recorded.	[19,20]
NCT02108002	1	Effect of Vorinostat on Nervous System Hemangioblastomas in Von Hippel-Lindau Disease (Missense Mutation Only)	Vorinostat	7	C Apr 2014	-	-
NCT02859441	1/2	A Phase I/II Trial for Intravitreous Treatment of Severe Ocular Von Hippel-Lindau Disease Using a Combination of the PDGF Antagonist E10030 and the VEGF Antagonist Ranibizumab	Ranibizumab & E10030 (anti-PDGF pegylated aptamer)	3	C Aug 2016	-	-
NCT03108066	2	PT2385 for the Treatment of Von Hippel-Lindau Disease-Associated Clear Cell Renal Cell Carcinoma	PT2385 Tablets (HIF-2α inhibitor)	4	O May 2017	-	-
NCT03401788	2	A Phase 2 Study of PT2977 for the Treatment of Von Hippel Lindau Disease-Associated Renal Cell Carcinoma	PT2977 (HIF-2α inhibitor)	61	C Mar 2018	27.9% showed partial response.	[34]

**Table 2 jcm-09-02740-t002:** Clinical monitoring of VHL patients with retinal HBs receiving propranolol in an off-label treatment: previous lesions, dosage, and response to the treatment.

Case	Gender, Age	Lesions Prior to Propranolol Treatment	Propranolol Treatment Initiation (months)	Propranolol Treatment Doses (mg/kg/day)	Response to Propranolol Treatment
1	Female, 32	Bilateral retinal HB	January 2017 (36 months)	Started at 0.66 and increased to 1.5	No lesions detected along the treatment
2	Male, 24	Medulla HB extracted and partial nephrectomy due to ccRCC left renal lesions	May 2018 (22 months)	Started at 0.66 and increased to 1.2	Stable bilateral cystic lesions after starting treatment
3	Male, 28	Left eye retinal HB, multiple tumoral and cystic lesions	August 2016 (47 months)	Started at 0.5 and increased to 1.4	Stable bilateral lesions but one after starting treatment
4	Male, 22	Right eye retinal HB and extracted cerebellum HB	April 2018 (16 months)	1.8	Stable 3 bilateral cystic lesions after starting treatment

**Table 3 jcm-09-02740-t003:** Primers used for q-PCR assays.

Gene	Sequence
***18S* Fwd** ***18S* Rev**	5′-CTCAACACGGGAAACCTCAC-3′ 5′-CGCTCCACCAACTAAGAACG-3′
***ADRB-1* Fwd** ***ADRB-1* Rev**	5′-GTGGAAGATGGGTGGGTTAG-3′ 5′-GAGCCACGATGATCGATTTTA-3′
***ADRB-2* Fwd** ***ADRB-2* Rev**	5′-CCATGTCCAGAACCTTAGCC-3′ 5′-GATCTGCGGAGTCCATGC -3′
***AQP-1* Fwd** ***AQP-1* Rev**	5′-GGAGGGTCCCGATGATCT-3′ 5′-CCTCCCTGACTGGGAACTC-3′
***BAX* Fwd** ***BAX* Rev**	5′-CACTCCCGCCACAAAGAT-3′ 5′-CAAGACCAGGGTGGTTGG-3′
***CAIX* Fwd** ***CAIX* Rev**	5′-TGCCGTCAATTAAGCATAAGG-3′ 5′-GTCCAGTAATCTGGGCAGGTA′-3
***CASP9* Fwd** ***CASP9* Rev**	5′-CCCAAGCTCTTTTTCATCCA-3′ 5′-TTACTGCCAGGGGACTCGT-3′
***IL-6* Fwd** ***IL-6* Rev**	5′-CAGGAGCCCAGCTATGAACT-3′ 5′-GAAGGCAGCAGGCAACAC-3′
***IL-1β* Fwd** ***IL-1β* Rev**	5′-CTGTCCTGCGTGTTGAAAGA-3′ 5′-TTGGGTAATTTTTGGGATCTACA-3′
***TNFAIP6* Fwd** ***TNFAIP6* Rev**	5′-GGCCATCTCGCAACTTACA-3′ 5′-GCAGCACAGACATGAAATCC-3′
***MDR* Fwd** ***MDR* Rev**	5′-TTGAAATGAAATGTTGTCTGG-3′ 5′-CAAAGAAACAACGGTTCGG-3′

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
