# Peer review of "Targeting β2-Adrenergic Receptors Shows Therapeutical Benefits in Clear Cell Renal Cell Carcinoma from Von Hippel–Lindau Disease"

_jcm, 2020, doi:10.3390/jcm9092740_

Round 1
Reviewer 1 Report
The authors have submitted a well-written and well organized manuscript to argue for the efficacy of ADRB antagonists (propranolol and ICI-118,551) against ccRCC in patients with VHL disease. The authors used VHL patient derived ccRCC cells in primary culture to analyze the effect of ADRB antagonists on survival. Using fluorescence based survival analysis, the authors found that human derived ccRCC cells were sensitive to propranolol and ICI (LD50 at 100µM) via induction of apoptosis. The authors then show that ADRB antagonists decrease HIF-2aplha expression in ccRCC (WB), VEGF secretion (ELISA on supernatant) and suppression of AQP1 expression (qrtPCR), a gene downstream in the HIF pathway. Analyzing HIF2alpha and p65 distribution in 786-O cells, the authors show reduced nuclear translocation. Using a scid mouse xenograft (786-O) model, the authors show that ADRB antagonists reduced tumor formation in two different experiments. Lastly, the authors review data on 4 patients that were treated with oral propranolol and report that the ccRCC tumors remained stable during propranolol administration.
The authors have tried to make a case for efficacy of ADRB antagonists against VHL related ccRCC. I can see the overarching storyline that the authors have created to support their case. However, the implementation needs to be improved. These are my specific major concerns:
- Consistency of experimental set-up: the critical flaw is the lack of consistent data using human ccRCC samples. Although, the initial studies were reported using human samples, the authors abandon these in favor of 786-O (a human VHL -/- ccRCC cell line). The major recommendation would be to perform and report experiments with available patient derived ccRCC cells for nuclear translocation studies and xenograft studies (PDX models).
- Another issue with consistency is that the authors have not reported how many human ccRCC tumors were used for the experiments. I would recommend the authors report on all of the experiments that were performed on any human ccRCC tumors that were procured for this study. Why weren’t all experiments performed on all of the ccRCC samples? Was it because there wasn’t enough tumor tissue? The risk with reporting partial data reporting is with suppression of unfavorable results.
- The clinical data reported in this manuscript are unconvincing at best. How much follow up did the patients have (in months)? How many lesions were measured? Were all lesions measured or were there some target lesions? What happened to renal cysts? How was dose escalation performed (timing)? Did propranolol result in flattening of the growth curve? What do the authors mean by ‘stable’ – zero growth or negligible growth? Were a-priori strategies used to define PR/CR/SD/PD such as in RECIST? The authors should report clearly the measurement strategies, blinding process, inter-observer variations and volumetric measurements (and trends).
Specific concerns:
- Table 1 is superfluous and does not add to the manuscript.
- Authors should report CAIX protein measurements (and not just qrtPCR).
- Authors need to present a comprehensive profile of HIF downstream gene expression (GLUT1/EPO/VEGF). A reporting of inter-sample differences in human ccRCC samples would be instructive.
- The authors report ADRB2 downregulation with propranolol and ICI. These results are intriguing and need to be explored and reported. These results are highly counter-intuitive. In many instances of receptor antagonism, gene upregulation is observed. Is this a transcriptional regulation? Is the gene promoter involved?
- Please explain how nuclear/cytoplasmic localization was measured?
Author Response
Please find attached the point by point answers

Reviewer 2 Report
This article focuses on targeting the beta 2 adrenergic receptors in therapy of clear cell renal cell carcinoma in patients with von Hippel-Lindau disease. The authors use various molecular and classic methods to provide insight into the potential mechanism of action and the efficacy of two ADRB antagonists in clear cell carcinoma growth suppression associated with the molecular role of the VHL protein function.
The authors were able to provide convincing evidence for the potential benefits of ADRB2 inhibition in patients with CC-RCC. I have found both the study design and methods to be sound, and the results stimulating. However, some small amendments might be necessary:
1) I find the general level of English to be appropriate, however some phrases require rewriting (e.g. 32-34). I also found the Introduction section to be quite wordy, including obvious or unnecessary information (e.g. 63-65, 82-84). In general, I would recommend to revise and abridge this section.
2) The Results sections would also benefit from some editing, and for the sake of clarity I would recommend to leave the comparison with the previous studies (e.g. 187-189) for the Discussion section, where they can be properly addressed, and potential explanation for the differing results given.
3) I would also like to point out that information about the informed consent is provided in the results section (120-121) and then repeated in materials and methods (551-553), I would suggest to remove it from the Results section completely.
4) Information about positive and negative control for the HIF-2 alpha as well as the concentrations used should be given in the Immunohistochemistry section (4.10.).
5) I would also recommend to provide explanations for the abbreviations (e.g. C,P,I) used in Figures 1, 2, 4 and 5.
Author Response
The point by point answer to reviewer 2 is attached

Round 2
Reviewer 1 Report
The authors have addressed all my concerns. I agree that at this time, the manuscript may need some more time before re-submission. This would be in order to address concerns specifically concerning CAIX WB and clinical tumor measurements. I do think that the requested changes will make this manuscript an important contribution to the field.
Author Response
See attached document
